# MeJA Elicitation of Chicory Hairy Roots Promotes Efficient Increase of 3,5-diCQA Accumulation, a Potent Antioxidant and Antibacterial Molecule

**DOI:** 10.3390/antibiotics9100659

**Published:** 2020-09-30

**Authors:** Guillaume Bernard, Harmony Alves Dos Santos, Audrey Etienne, Jennifer Samaillie, Christel Neut, Sevser Sahpaz, Jean-Louis Hilbert, David Gagneul, Nathalie Jullian, Ali Tahrioui, Sylvie Chevalier, Céline Rivière, Caroline Rambaud

**Affiliations:** 1BioEcoAgro, Joint Research Unit 1158, Univ. Lille, INRAE, Univ. Liège, UPJV, YNCREA, Univ. Artois, Univ. Littoral Côte d’Opale, ICV—Institut Charles Viollette, F-59650 Villeneuve d’Ascq, France; guillaume.bernard04000@gmail.com (G.B.); harmony.alves-dos-santos@univ-lille.fr (H.A.D.S.); audrey.etienne@univ-lille.fr (A.E.); jennifer.samaillie@univ-lille.fr (J.S.); sevser.sahpaz@univ-lille.fr (S.S.); jean-louis.hilbert@univ-lille.fr (J.-L.H.); david.gagneul@univ-lille.fr (D.G.); nathalie.pawlicki@u-picardie.fr (N.J.); celine.riviere@univ-lille.fr (C.R.); 2U1286 Infinite, University of Lille, INSERM, CHU Lille, 59000 Lille, France; christel.neut@univ-lille.fr; 3EA4312 Laboratoire de Microbiologie Signaux et Microenvironnement, Univ. de Rouen Normandie, 27000 Évreux, France; ali.tahrioui@univ-rouen.fr (A.T.); sylvie.chevalier@univ-rouen.fr (S.C.)

**Keywords:** hairy roots, *Cichorium intybus*, 3,5-dicaffeoylquinic acid 3,4,5-tricaffeoylquinic acid, *Pseudomonas aeruginosa*, pyoverdine

## Abstract

*Cichorium intybus* L. (*Asteraceae*) is an important industrial crop, as well as a medicinal plant which produces some bioactive compounds implicated in various biological effects with potential applications in human health. Particularly, roots produce hydroxycinnamic acids like 5-caffeoyquinic acid and 3,5-dicaffeoylquinic acid (di-CQA). The present investigation relates to the use of methyl jasmonate for enhancing phenolic compounds accumulation and production in hairy root cultures of *C. intybus*. Elicitated hairy root growth rate increased 13.3 times compared with the initial inoculum in a period of 14 days and di-CQA production represented about 12% of DW. The elicitation has also promoted the production of tricaffeoylquinic acid never described in the chicory roots and identified as 3,4,5-tricaffeoyquinic acid by means of nuclear magnetic resonance. Our study confirmed the strong anti-oxidant effect of di-CQA. Our results also confirmed globally a selectivity of action of di-CQA against Gram-positive bacteria, in particular against some strains of *Staphylococcus* and *Streptococcus*. However, a non-negligible antibacterial activity of di-CQA against *Pseudomonas aeruginosa* was also underlined (MIC = 0.156 mg.mL^−1^ against some *P. aeruginosa* strains). The influence of di-CQA has been explored to evaluate its impact on the physiology of *P. aeruginosa*. Di-CQA showed no effect on the biofilm formation and the production of extracellular pyocyanin. However, it demonstrated an effect on virulence through the production of pyoverdine with a dose-dependent manner by more than 7-fold when treated at a concentration of 128 µg·mL^−1^, thus suggesting a link between di-CQA and iron sequestration. This study shows that elicitated hairy root cultures of chicory can be developed for the production of di-CQA, a secondary metabolite with high antibacterial potential.

## 1. Introduction

Phenolics are a group of specialized metabolites synthesized by plants for protection against abiotic stressors such as UV radiation [1] and for competitive warfare against insects, viruses, microorganisms and other plants [2]. They are also responsible for the color, smell and flavor of plants [3]. They mainly arise from the phenylpropanoid pathway. Among them, hydroxycinnamic acids (i.e., caffeic, *p*-coumaric, sinapic and ferulic acids) consisting of a C_6_ aromatic ring linked to a C_3_ aliphatic side chain (C_6_–C_3_) are widely distributed in the plant kingdom. They are rarely accumulated under their free forms and are usually conjugated with acids, amines or glucosides to form esters, phenolamides or glycosides, respectively. Chlorogenic acid (also known as 3-caffeoylquinic acid), obtained by esterification of caffeic acid and L-quinic acid, is widely distributed in plants and is one of the most representative hydroxycinnamic acid derivatives. It is highly accumulated in different families including the Solanaceae, Rubiaceae and Asteraceae, such as artichoke (*Cynara scolymus*), yarrow (*Achillea millefolium*), tarragon (*Artemisia dranunculus L.)* or chicory (*Cichorium intybus* L.) [4,5].

*Cichorium intybus* L., known and used as a medicinal plant in many regions of the world, could be divided into four “varieties” according to its use: (i) industrial chicory used for the extraction of inulin and the production of a substitute of coffee; (ii) forage chicory used as a component in grazing pastures because of its resistance to drought and its high nutritional value; (iii) the witloof chicory commonly cultivated in Europe by forcing to produce etiolated edible buds known as “chicon”; (iv) leaf chicory whose leaves are eaten raw or cooked [6]. *Cichorium intybus* L. has numerous biological activities, including antimicrobial, anthelmintic, antimalarial, hepatoprotective, antidiabetic, gastroprotective, anti-inflammatory, analgesic, antioxidant, tumor-inhibitory and antiallergic activities [7,8]. These biological properties could be attributed to inulin, vitamins, and specialized metabolites such as sesquiterpene lactones, flavonoids, coumarins, hydroxycinnamic acids as well as alkaloids that are present in the different parts of *C. intybus* L. [9].

*C. intybus* L., like other plants of the Asteraceae family, is particularly rich in hydroxycinnamic acid derivatives such as 3-caffeoylquinic acid (CQA) and 3,5-dicaffeoylquinic acid (di-CQA). In addition, caftaric and chicoric acids, two esters of tartaric acid, are also accumulated in *C. intybus* L. [5,10,11]. Interestingly, an original tissue distribution of these esters was highlighted. Tartaric esters were shown to be mainly accumulated in aerial part [12] whereas di-CQA was mainly located in the root parts [5]. The distribution of CQA was uniform between these organs. Multiple biological roles have been attributed to these caffeoyl ester derivatives. For example, tomatoes engineered to accumulate increased levels of CQA were shown to be more resistant to microbes [13]. CQA is also a key intermediate in the synthesis of lignin [14]. Considering the putative health benefits of di-CQA, in particular antioxidant [15,16], anti-inflammatory [17] and antimicrobial activities [16], it is tempting to search for alternative way of supply even if it is present at high concentrations in root tissue [5]. Therefore, it is important to consider biotechnological methods for the production of di-CQA.

Hairy root culture (HRC) is a unique tool for the synthesis of high-value specialized metabolites. HRC has the same capacity for biosynthesis and accumulation, or even more, of specialized metabolites compared to the mother plant and is a fast-growing tissue. HRC can be grown in hormone-free media and are therefore genetically stable. HRC also allows the production of metabolites that are not synthesized by the mother plant [18,19]. Interestingly, the production of metabolites using this alternative method can be improved by elicitation. Specialized metabolites can be secreted and trapped in the medium which facilitates their extraction and purification [20]. Several studies have shown that chicory HRCs can be obtained easily and produced a huge biomass [21,22]. The HRC of chicory have been successfully used for the production of coumarins [23], sesquiterpene lactones [24,25] and CQA derivatives in large quantities [24,26] among which di-CQA was the main compound.

This study aims to evaluate the capacity of chicory HRC under elicitation by methyl jasmonate (MeJA), to produce large quantities of caffeoylquinic acid derivatives, which are of interest for the development of biologically active compounds. The antibacterial and antioxidant activities of chicory hairy root extracts, sub-extracts as well as their main constituents were also evaluated. In addition, the effect of di-CQA on the physiology of *Pseudomonas aeruginosa* was evaluated by studying its impact on the production of virulence factors and the formation of biofilm.

## 2. Results

### 2.1. Establishment of the HRCs and Growth Parameters in Liquid Medium

The infection of chicory leaves (*C. intybus* L.) with *Rhizobium rhizogenes*, allowed the emergence of roots in the wounds made with the scalpel after approximately two weeks of culture on medium containing ampicillin. Many roots were harvested and subsequently grown independently in MS × 0.5 medium. They were then considered as individual HR lines and evaluated for their ability to produce biomass and hydroxycinnamic acids. Among them, one HR line ‘Orchies 2659-31’ was selected because of its capacity to produce biomass and phenolic compounds compared to the other lines (not shown). After five passages on medium with ampicillin and one passage without ampicillin, PCR amplification of *rolB* and negative PCR of *virD2* confirmed the successful transformation and elimination of *R. rhizogenes* from the chicory hairy root (data not shown). In order to determine the optimal moment for elicitation, the culture parameters were determined by establishing the growth kinetics of the HR line ‘Orchies 2659-31’. The growth phase lasts 13–14 days and the maximum biomass obtained on day 15 was approximately 4 g in 20 mL of culture medium, which represents 200 g FW.L^−1^ (Figure 1). The growth Index (GI) was about 13 after 14 days of culture.

In addition, to determine whether the roots can grow in larger volumes, a scale-up was performed in 20 mL, 40 mL, 100 mL, 200 mL and 400 mL of liquid medium inoculated with 300 mg, 600 mg, 1.5 g, 3.0 g and 6.0 g of FW, respectively. Figure 2 shows that the chicory HRC can grow in 400 mL of medium and produce proportionally the same amounts of biomass as in smaller volumes. To obtain this quantity of biomass, the speed of the stirring was limited to 90 rpm for the flasks containing 100 mL and 200 mL of culture medium and to 80 rpm for the flasks containing 400 mL of culture medium. Our HR line produced 200 g L^−1^ FW within 14 days and the GI was 13.3, a higher rate and a better GI than those obtained in early study on chicory [24] in which the GI was 10 and using a different combination of chicory variety and *Rhizobium rhizogenes* strain. In our study, the biomass was measured in small flasks of 50 mL with only 20 mL of culture medium but scale up study from 20 mL until 400 mL did not show any difference in the production of biomass (Figure 2).

### 2.2. Improvement of Specialized Metabolite Production

The chicory hairy roots mainly produce caffeoylquinic acids such as 3-caffeoylquinic acid (CQA; chlorogenic acid) and 3,5-dicaffeoylquinic acid (di-CQA; isochlorogenic acid A) [24]. The chicory hairy root line ‘Orchies-2659-31’ was able to produce approximately 1% DW of CQA and 5% DW of di-CQA (Figure 3), which represents 60 mg L^−1^ of CQA and 300 mg L^−1^ of di-CQA. The production of these specialized metabolites was increased under elicitation with different concentrations of methyl jasmonate (MeJA) during the evaluation at 6, 9 and 12 days after inoculation (Figure 3).

The elicitation experiments were particularly efficient for di-CQA synthesis and accumulation. After 6 days of growth with MeJA, the chicory hairy roots were able to produce approximately 10% of di-CQA (from DW), which represents a 100% increase in the production of di-CQA as compared to the control condition (without the elicitor MeJA). Nine days and 12 days after elicitation with 0.3 and 0.45 mM of MeJA, respectively, the production of di-CQA decreased slightly but in the presence of 0.15 mM of MeJA there was still a small increase in the production of di-CQA. The di-CQA maximal production (12% of the DW) was reached after 12 days under elicitation with 0.15 mM MeJA, which represents 720 mg L^−1^. A decrease in the biomass was not observed with 0.15 mM of MeJA. Interestingly, under elicitation with MeJA, chromatogram analysis showed the presence of a new compound with a retention time of 12 min and a UV spectra characteristic of chlorogenic acid compounds (Figure 4). After mass spectrometry analysis this compound showed a *m/z* ratio at 677.15 which could correspond to an isomer of tri-caffeoylquinic acid.

### 2.3. Tri-Caffeoylquinic Acid Production in Chicory Hairy Roots

The production of tri-caffeoylquinic acid was evaluated as a function of the MeJA concentration and the elicitation time (6, 9, and 12 days), then the relative absorbance of the peaks was analyzed. The tri-CQA content of hairy roots increased with the duration of elicitation and with the concentration of MeJA (Figure 5).

Notably, 0.3 mM and 0.45 mM MeJA were the optimal concentrations for the accumulation of tri-CQA as compared to the non-elicitated chicory hairy roots. The production of tri-CQA increased by about 7-fold. However, after 3 days with MeJA concentrations up to 0.15 mM, MeJA has an effect on the growth of hairy roots and induces growth inhibition and necrosis, that has an impact on the production of biomass and so on the quantity of tri-CQA produced by flask (Figure 6). But when the tri-CQA rate per flask was calculated, 0.45 mM was always the best concentration and 6 days were enough to obtain the maximum rate. In this case, the increase was only 5-fold.

Interestingly, elicitation of chicory hairy roots with MeJA allowed the production of tri-CQA metabolite that was not accumulated in non-elicitated hairy roots. Our analysis showed that the concentration of elicitor and the exposure time strongly influence the synthesis of specialized metabolites, since two different concentrations of MeJA could have a different effect on the accumulation of CQAs. Although high concentrations of MeJA have a small necrotic effect on hairy roots, the rates of CQAs accumulation are still considerable.

### 2.4. Purification and Structural Identification of the Major Isomers of di-CQA and tri-CQA

The two major isomers of di-CQA and tri-CQA were purified by preparative HPLC from the ethyl acetate (EtOAc) sub-extract of the elicitated *C. intybus* L. hairy roots, after optimization of the gradient by HPLC-UV (Appendix A). The yield obtained for the major isomer of di-CQA was 10 times higher than the yield obtained for the major isomer of tri-CQA.

After the purification process, the purity of both isomers was checked by UHPLC-UV-MS. The identification of 3,5-dicaffeoylquinic acid (di-CQA) and 3,4,5-tricaffeyolquinic acid (tri-CQA) was established on the basis of the comparison of the 1D- and 2D-NMR data of the two purified major isomers (compounds **1** and **2**) with those reported in the literature [27,28] (Figure 7).

### 2.5. Antimicrobial Activity and CQAs Quantification

The antimicrobial activity of the crude methanolic extract obtained from a non-elicitated (HR1) and an elicitated (HR2) *C. intybus* L. hairy roots was evaluated towards several bacterial clinical isolates (Gram-positive and Gram-negative) and two *Candida* strains. The two crude methanolic extracts were weakly active or in most of cases not active (MIC > 1.25 mg·mL^−1^). The crude methanolic extract HR2 was significantly more active than the crude methanolic extract HR1, with MIC values of 0.625 mg·mL^−1^ against some strains of *Staphylococcus* (data not shown).

A liquid/liquid partition of the crude methanolic extracts with ethyl acetate and water was performed to concentrate the isomers of the di-CQAs and the isomers of the tri-CQAs (Appendix A). The ethyl acetate sub-extract, obtained in a low yield in comparison with the aqueous sub-extract, was particularly enriched in di-CQA and tri-CQA (but not in CQA) (Appendix A). The aqueous sub-extract contained CQA, a low quantity of di-CQA and other metabolites of unknown nature. The tri-QCA metabolite was not detected in the aqueous sub-extract (Appendix A). Quantification of CQA, di-CQA and tri-CQA was reported in Table 1.

We next performed a second screening for the antimicrobial activity of ethyl acetate and aqueous sub-extracts against the same panel of microorganisms. The antimicrobial activity of two pure compounds, CQA and di-CQA, was also evaluated. The small amount of tri-CQA obtained after purification did not allow us to test the antimicrobial activity of this metabolite. The two aqueous sub-extracts (HR1 and HR2) were inactive against all the strains tested (data not shown, MIC > 1.25 mg·mL^−1^) in comparison with the ethyl acetate sub-extracts (Table 2).

A selectivity of action of hairy roots ethyl acetate sub-extracts and caffeoylquinic acids has been observed against Gram-positive bacteria, in particular against some strains of *Staphylococcus* and *Streptococcus*. However, a non-negligible antibacterial activity is also to be emphasized on certain strains of Gram-negative bacteria, such as *Proteus* and *Pseudomonas*. This broad spectrum of action is particularly interesting. However, the ethyl acetate sub-extracts and the two caffeoylquinic acids showed no activity on the *Enterococcus* strains (Gram positive), as well as on the strains of *Enterobacter*, *Escherichia coli* and *Klebsiella pneumoniae* (Gram negative). The compounds CQA and di-CQA globally have the same antibacterial activity with the same selectivity of action. Their antimicrobial activity is particularly interesting with regard to the strains of *Staphylococcus* and *Streptococcus* (frequent in skin infection) (MIC = 0.156 mg.mL^−1^), as well as with respect to the strains of *Pseudomonas aeruginosa* (MIC = 0.156 mg.mL^−1^ for di-CQA) and the strains of *Candida albicans* (MIC = 0.078 mg.mL^−1^) (both important species in nosocomial infections difficult to treat) (Table 2).

Since the aqueous sub-extracts were not active, it can be assumed that the levels of CQA in the aqueous sub-extracts were quite low compared to other metabolites without antibacterial activity or that other metabolites antagonized its activity. Similarly, the levels of CQA and di-CQA in the crude extract were probably quite low relative compared to other metabolites or their effect was antagonized.

### 2.6. Effect of 3,5-Dicaffeoylquinic Acid on Pseudomonas aeruginosa Virulence Factors Production and Biofilm Formation

The effect of di-CQA against *P. aeruginosa* was evaluated at concentrations ranging from 2 to 256 µg·mL^−1^, in a liquid medium with the model strain *P. aeruginosa* H103. The di-CQA compound shows no effect on the biofilm formation (Figure 8A) and the production of extracellular pyocyanin (Figure 8B). However, it increases the production of pyoverdine in a dose-dependent manner by more than 7-fold when treated at a concentration of 128 µg·mL^−1^ (Figure 8C). Since the amount of pyoverdine increased in response to di-CQA treatment, our data suggest that this compound would interact with Fe^3+^.

Pyoverdine is a Fe^3+^ high-affinity siderophore that is produced by *P. aeruginosa* specifically in response to medium iron starvation. Since pyoverdine production was increased in response to 3,5-dicaffeoylquinic acid (di-CQA) treatment, we speculated the existence of a link between iron starvation and di-CQA. Interestingly, 3-caffeoylquinic acid (CQA) also known as chlorogenic acid was previously shown to chelate iron (Fe^3+)^ in a ratio of 3/1 (chlorogenic acid/iron) in vitro at pH 7.4 [29]. Moreover, it has been reported that the antioxidant effect of CQA is attributed to the chelate structure with iron [29,30]. Thus, our data suggest that 3,5-dicaffeoylquinic acid (di-CQA) as being a derivative of 3-caffeoylquinic acid might also sequester iron, limiting thus the Fe^3+^ availability, which in turn would result in increased pyoverdine siderophore production by *P. aeruginosa.* However, this hypothesis deserves further research.

### 2.7. Antioxidant Activity

The antioxidant activity of extracts and sub-extracts of *C. intybus* L. hairy roots elicitated and non-elicitated, and of pure compounds CQA and di-CQA was evaluated using the DPPH test. *C. intybus* L. elicitated hairy roots (HR2; IC_50_ = 52.76 µg·mL^−1^) demonstrated a higher DPPH radical-scavenging activity than non-elicitated hairy roots (HR1; IC_50_ = 89.96 µg·mL^−1^) (Table 3). These results suggest that the elicitation with MeJA increases the antioxidant activity of the hairy roots. Thus, the antioxidant potential of di-CQA, the main compound isolated from *C. intybus* L. hairy roots, has been shown to be higher than that of CQA. Accordingly, a stronger radical scavenging activity of di-CQA compared to CQA and tri-CQA, was previously underlined in [16]. Remarkably, the ethyl acetate sub-extracts of both HR1 and HR2 showed a strong DPPH radical-scavenging, which is higher than that of their corresponding crude methanolic extracts. This increase in antioxidant activity can probably be due to the fact that di-CQA has been concentrated in these extracts. Together, these data demonstrated the high radical scavenging activity of the ethyl acetate sub-extracts enriched in di-CQA and confirmed the highest antioxidant activity of di-CQA in comparison to CQA.

## 3. Discussion

The hairy root lines of *C. intybus* L. (chicory) produce CQAs, mainly CQA and di-CQA. If we compare the biomass, the culture time and the content of CQA and di-CQA, the hairy root line of ‘Orchies 2659-31’ is interesting because it grows faster than other HRCs. Remarkably, the accumulation of tri-CQA in chicory hairy root lines has not been observed previously. No other species have been able to provide such high yields of CQAs in such a short time. Yet HRCs of other species of the Asteraceae family have also been investigated including *Echinacea purpurea* [31], *Lactuca virosa* [32], *Rhaponticum carthamoides* [33], *Sphagneticola calendulacea* [34], *Aster scaber* [35] or *Ligularia fischeri* [36] but the CQAs content/biomass rates are always lower than those obtained in *C. intybus* L. HRCs, in particular the content in di-CQA, which is high in *C. intybus* L. Only the hairy root lines of *Lactuca virosa* could be competitive with the hairy root lines of *C. intybus* L. because they have a GI of 16, but maximum growth is obtained within 30 days of culture, against 14 days for chicory and these cultures produce the same compounds as chicory (CQA and di-CQA), a little more CQA than the hairy roots of chicory but less di-CQA, and as the di-CQA is more antioxidant and antimicrobial than CQA, the chicory HRCs are more interesting. In the other species, the GI was close to that of chicory but the growth was always slower and the contents of CQA and di-CQA were lower. In *R. carthamoides*, the GI was 12.4, close to that of *C. intybus* L., but the highest accumulation of hairy root biomass was reached after 35 days of culture. *R. carthamoides* is rich in CQAs, in particular tri-CQA derivatives, but the total CQAs content is much lower than that of *C. intybus* L. Other species, *E. purpurea*, *S. calendulacea*, *A. scaber*, and *L. fischeri* are not competitive with chicory because they accumulate very small amounts of CQAs, mainly chlorogenic acid, or not at all from CQAs, like *S. calendulacea*.

Elicitation is a promising method to increase the production of phenolic compounds. In the present study, elicitation with MeJA was used in *C. intybus* L. HRCs. MeJA is a well-known elicitor and has been reported as a signal transduction elicitor for the plant defense response and the production of plant secondary metabolites [37,38]. Therefore, the MeJA elicitor allowed to increase the rate of CQAs accumulation, in particular the production of di-CQA was enhanced by about 3 times. Similar results have been obtained in HRCs of *Aster scaber* [35] in which, the yields of chlorogenic acid were doubled with MeJA used as an elicitor. Similarly, in HRC of *Ficus carica* grown in the presence of MeJA, the production of CQA was increased by 4.4-fold [39]. Other elicitors such as salicylic acid (SA), yeast extract (YE) or cyclodextrin (CD) could be used but MeJA is a more general inducer of the production of specialized metabolites than the others, in particular on the metabolism of phenolic compounds. The effect of MeJA and SA was compared on chicory HRCs [40], in which sesquiterpene lactones were analyzed. The results of this study [40] showed that SA is a better inducer of sesquiterpenes metabolism than MeJA, which was not able to increase the production of sesquiterpene lactones.

According to our results, the ethyl acetate sub-extract from the elicitated hairy roots (HR2) was slightly more active than the ethyl acetate sub-extract of the non-elicitated hairy roots (HR1), in particular against some *Staphylococcus* strains. This could be explained by the higher levels of di-CQAs and tri-CQAs in the elicitated hairy roots. The antimicrobial potential of caffeoylquinic acids has indeed already been demonstrated against Gram-positive bacteria, in particular against *Staphylococcus* strains [16,41,42,43]. A recent study [16] showed that tri-CQA was more active than di-CQA against penicillin sensitive and resistant *S. aureus* strains, as well as methicillin-resistant *S. aureus*. While the effect of CQA on the virulence factors and pathogenicity of *P. aeruginosa* is well understood [44,45], the antibacterial potential of di-CQA against *P. aeruginosa* strains is controversial and requires further investigation. A number of studies have shown a lack or a weak antibacterial activity of this metabolite against *P. aeruginosa* [46,47], however, others studies have found approximately MIC values similar to ours of di-CQA against *P. aeruginosa*. It is the case with the study of Venditti et al. [48] but merely the study of Lehbili et al. [49] also using a multiple inoculator and indicating a MIC value of 125 µg·mL^−1^, but results may be strain-dependent.

To evaluate the impact of di-CQA on the physiology of *P. aeruginosa*, its effect on the biofilm formation as well as its effect on virulence through the production of pyocyanin and pyoverdine were studied. *P. aeruginosa* is an environmental bacterium, which is well-known due to its huge adaptation abilities to many environments including water, soils, plants, nematodes, or animals including humans [50]. It is also an opportunistic pathogen involved in numerous chronic and acute life-threatening infections, which are closely related to its sessile and free-living lifestyles, respectively [51]. In chronic infections, *P. aeruginosa* develops a community, in which bacteria are embedded into an exoproduct matrix, while in acute infections, *P. aeruginosa* is planktonic and deploys a collection of virulence factors, among which pyocyanin and pyoverdine [52]. Pyocyanin, a phenazine-derived blue-green pigment is a redox-active secondary metabolite promoting the colonization and the dissemination of the bacterium. The green fluorescent pigment pyoverdine is a Fe^3+^ high affinity siderophore [53]. Iron homeostasis plays a central role in bacterial growth and survival, and *P. aeruginosa* has developed several strategies to chelate iron from the available environmental iron sources, among which the catechol-derived siderophore pyoverdine. In response to iron starvation, *P. aeruginosa* upregulates siderophore biosynthesis and iron-trafficking pathways. Remarkably, this restricted iron availability also acts as an environmental signal to regulate the expression of other genes *in vivo,* notably virulence factors, such as exotoxins. Altogether, our data show that di-CQA did not affect biofilm formation or pyocyanin production under our conditions, but does result in a strong response to iron deprivation, suggesting a link between this compound and the iron chelation.

## 4. Materials and Methods

### 4.1. Plant Material and Rhizobium Strain

Seeds of industrial chicory, *Cichorium intybus* L. var Orchies (Florimond Desprez SA, Cappelle en Pévèle, France) were sterilized with 0.1% HgCl_2_ for 20 min and germinated in Heller (1953) culture medium [54] supplemented with 20 g L^−1^ sucrose and 6 g L^−1^ agar, pH 5.6. Young leaves of 12 days-old seedlings were used for *Rhizobium* (Agrobacterium) infection. *Rhizobium rhizogenes* strain 2659 (kindly provided by Marc BUEE, INRAE Nancy, France), a cucumopine-type strain, was used for plant transformation. Strains were stored in glycerol at −80 °C and plated on YEB medium [55] for three days at 28 °C in the dark.

### 4.2. Establishment of HRC and Molecular Confirmation of Their Phenotype

Three-day-old bacterial colonies were used to infect the leaves. A scalpel was dipped in the colonies, then incisions were made in the leaves at the ribs with the contaminated scalpel. Infected leaves were plated on a solid Murashige and Skoog (MS) medium [56] with 30 g·L^−1^ sucrose and 6 g· L^−1^ agar, pH 5.6, and cultured for three days at 22 °C, under a 16 h light photoperiod provided by cool-white fluorescent lamps (40 µmol·m^−2^·s^−1^). The infected leaves were then transferred to a new solid MS culture medium containing 300 mg·L^−1^ ampicillin to remove remaining *Rhizobium* and were cultured under the same conditions as before, and transferred again as soon as necessary. Roots emerging from the leaves, identified as independent transformation events, were picked and cultured separately in solid MS × 0.5 medium (MS salts × 0.5, vitamins × 0.5, Fe-EDTA × 0.5, 30 g·L^−1^ sucrose, pH 5.6) culture medium with antibiotics. After three cycles of subculture on a solid medium containing antibiotics, the bacteria-free hairy roots (HRs) were transferred to a liquid MS × 0.5 medium without antibiotics.

DNA extraction from roots of vitroplants (negative control) and HR clones was performed using the Nucleospin DNA Plant Mini Kit (Macherey-Nagel, Düren, Germany). DNA from *R. rhizogenes* strain 2659 was used as a positive control. Subsequently, the PCR amplification of the genes virD2 and rolB using specific primers (virD2 F: 5′-ATG-CCC-GAT-CGA-GCT-CAA-GT-3′; R: 5′-CCT-GAC-CCA-AAC-ATC-TCG-GCT-GCC-CA-3′) [57] and rolB F: 5′-CCT-CCC-TGC-CGT-ACA-CAT-TT-3′; R: 5′-CAC-AAA-AGC-CTG-GAA-CCA-GC-3′) [58] was carried out to confirm the transformed nature of the HRs and the elimination of bacteria. PCR amplification was carried out in a 15 μL reaction volume containing 2 µL DNA solution, 1 µL of each primer (1.25 µM), 1.5 µL of 10× Taq polymerase buffer, 1.2 µL MgCl_2_ (25 mM, Roche, Basel, Switzerland), 0.6 µL dNTP mix (2.5 mM, Applied Biosystems, Foster City, Calif., USA) and 1.5 µL of Taq polymerase (5 U·µL^−1^, Applied Biosystems). PCR reaction was performed at an initial denaturation temperature of 94 °C for 4 min, followed by 35 cycles of 94 °C for 30 s, 58 °C for 1 min and 72 °C for 1 min, and a final step of 10 min at 72 °C. The PCR products were separated using electrophoresis on 1% agarose gel in TAE 0.5× buffer and visualized under UV light after staining with GelRed^TM^ (Biotium, San Francisco, CA, USA).

### 4.3. Measurement of HRC Growth

Twenty mL of culture medium was inoculated with 300 mg of HRCs, and samples were harvested, every 3 or 5 days, by vacuum filtration using a Büchner funnel lined with Whatman filter paper and washed with distilled water. Fresh weight was recorded and the plant material was immediately frozen in liquid nitrogen and stored at −80 °C before lyophilization. After freeze-drying, dry weight was measured. Each point on the curves represents the mean of three independent determinations (flasks). The growth index (GI) was calculated using the ratio of final fresh weight to initial fresh weight.

### 4.4. Elicitation and Scale Up

Elicitation was performed in triplicate using 12-day-old HRCs. Twenty mL of culture medium was inoculated with 300 mg of roots and methyl jasmonate (MeJA) was applied as an elicitor after 12 days of culture. A stock solution of MeJA (40 mM) was prepared in 100% (*v/v*) ethanol, then filter-sterilized and added to the 12-day-old root cultures at a final level of concentration of 0.1 mM, 0.15 mM, 0.25 mM, 0.5 mM or 0.75 mM. Cultures without ethanol or MeJA, on the one hand, and without MeJA but with ethanol (25 µL/20 mL), on the other hand, were used as controls. The roots were then collected after 3, 6, 9 and 12 days. The biomass (fresh and dry weight) and the content of phenolic acids were determined.

Scaling has been implemented to obtain larger quantities of biomass. HRCs were made in Erlenmeyer flasks of different sizes and in different volumes of culture medium. Five sizes of Erlenmeyer flasks (50 mL, 100 mL, 250 mL, 500 mL and 1 L) filled with 20 mL, 40 mL, 100 mL, 200 mL or 400 mL of culture medium, respectively, were tested. The biomass and the content of phenolic compounds were determined after 14 days of culture.

### 4.5. Extraction and Analysis of Polyphenols

The lyophilized roots (approximately 25 mg) were ground (1 min at 30 Hz) using a grinder (Retsch, Haan, Germany). Then 1 mL of extraction buffer (75% methanol, 23% water, 2% acetic acid) was added to the powder and the mixtures were incubated at 4 °C in a gyratory shaker at 40 rpm, overnight. After centrifugation (15 min at 14,000× *g*), the supernatant was recovered and 1 mL of extraction buffer was added to the pellet. After centrifugation at 14,000× *g* for 15 min, the supernatants were combined and stored at −20 °C before being analyzed. Metabolite analysis was carried out using a Prominence HPLC system (Shimadzu, Marne-la-Vallée, France) consisting of a quaternary pump (LC-20AD) and a UV-visible diode array detector (SPD-20A). The chromatographic procedure was as described in [59] for HR.

### 4.6. Extraction and Purification of 3,5-Dicaffeoylquinic Acid (di-CQA) and 3,4,5-Tricaffeoylquinic Acid (tri-CQA)

The crude hydromethanolic extract was obtained after a methanol-water (75:25) mixture-based extraction (800 mL) of the chicory freeze-dried HRCs (18.3 g) with three successive macerations at room temperature. After filtration and evaporation of methanol, the filtrate was frozen and then freeze-dried. The percentage yield on the basis of the dry weight of crude extract was 58% (10.57 g of crude extract). Then, the crude extract (10.28 g) was further fractionated by a liquid-liquid separation using ethyl acetate (300 mL) and water (300 mL), repeated four times successively. The corresponding sub-extracts were freeze-dried and yield percentages of 11% and 73% respectively were obtained.

High Performance Liquid Chromatography (HPLC)-UV analyzes were carried out using a Shimadzu^®^ system, with two LC-10AS pumps, a SCL-10A controller and a SPD-M20A diode array detector. LabSolution software (version 5.87) was used. The stationary phase was an Uptisphere strategy (Interchim^®^, Montluçon, France). C_18_-HQ-250 (5 µm, 250 × 4.6 mm) column. The mobile phase was composed of 0.1% of formic acid in water (solvent A) and acetonitrile (solvent B). After optimization, the following proportions of solvent B were selected: 10–20% (0–5 min), 20–25% (5–30 min), 25–30% (30–35 min), 30–40% (35–40 min), 40–100% (40–42 min) and 100% (42–50 min) at 1 mL.min^−1^. Twenty µL of a solution of 10 mg.mL^−1^ ethyl acetate sub-extract in methanol were injected after filtration (PTFE filter 0.45 µm). Phenolic compounds were detected at 320 nm (maximum absorbance of caffeoylquinic acids).

The preparative HPLC was performed using a Shimadzu^®^ HPLC system equipped with a LC-20AP binary high-pressure pump, a CBM-20A controller, and a SPD-M20A diode array detector. The software used was LabSolution (version 5.87). The stationary phase was an Utisphere Strategy SUM US5 C18HQ-250/212 (5 µm, 250 × 21.2 mm) (Interchim^®^, Montluçon, France). The mobile phase was composed of water (solvent A) and acetonitrile (solvent B). The following proportions of solvent B were used: 10–20% (0–5 min), 20–25% (5–30 min), 25–30% (30–35 min), 30–40% (35–40 min), 40–100% (40–42 min) and 100% (42–50 min) at 16 mL.min^−1^. Injections with 500 µL of a solution of ethyl acetate sub-extract at 80 mg.mL^−1^ in methanol (160 mg in four injections) were performed, after filtration (PTFE filter 0.45 µm). Phenolic compounds were detected at 320 nm (maximum absorbance of caffeoylquinic acids). Fifty mg of di-CQA (31.25%) and 5 mg of tri-CQA (3.1%) were obtained.

### 4.7. UHPLC-UV-MS Analyzes and Structural Identification by NMR

Ultra-High Performance Liquid Chromatography (UHPLC) analyzes were carried out using an Acquity UPLC^®^ H-Class Waters^®^ system (Waters, Guyancourt, France) equipped with a diode array detector (DAD) and an Acquity QDa ESI-Quadrupole Mass Spectrometer. The software used was Empower 3. The stationary phase was a Waters^®^ Acquity BEH C18 column (2.1 × 50 mm, 1.7 μm) connected to a 0.2 µm in-line filter. The mobile phase consisted of 0.1% formic acid in water and 0.1% formic acid in acetonitrile. The gradient of acetonitrile was 10–20% (0–1 min), 20–30% (1–6 min), 30% (6–7 min), 30–40% (7–8 min), 40–100% (8–8.5 min) and 100% (8.5–10 min) at 0.3 mL.min^−1^. The temperature of the column was adjusted to 30 °C. Solutions of extracts or compounds were prepared in MeOH at 100 µg·mL^−1^. The injection volume was 2 µL. Phenolic compounds were detected at 320 nm (maximum absorbance of caffeoylquinic acids).

The ionisation was performed in negative mode. The cone voltage was set at 10 V. The temperature of the probe was 600 °C. The capillary voltage was 0.8 kV. The MS-Scan mode was used from 100 to 1000 Da. The purified compounds were identified using Nuclear Magnetic Resonance (NMR). NMR spectra were recorded on a Bruker DPX-500 spectrometer. Mono- (^1^H and ^13^C) and bi-dimensional (COSY, HSQC, HMBC) spectra were carried out for the two major purified isomers of dicaffeoylquinic acid and tricaffeoylquinic acid. These compounds were solubilized in deuterated methanol (MeOD).

*3,5-Dicaffeoylquinic acid*: White powder (50 mg); ESI-MS *m/z* 515.32 [M–H]^−^ (C_25_H_24_O_12_ requires 516); ^1^H-NMR (500 MHz, MeOD): 𝛿 7.64 (1H, *d*, *J* = 15.9 Hz, H-7′), 7.60 (1H, *d*, *J* = 15.9 Hz, H-7″), 7.09 (1H, *d*, *J* = 2.1 Hz, H-2′), 7.09 (1H, *d*, *J* = 2.1 Hz, H-2″), 6.99 (1H, *dd*, *J* = 7.9, 2.1 Hz, H-6″), 6.98 (1H, *dd*, *J* = 7.9, 2.1 Hz, H-6′), 6.8 (1H, *d*, *J* = 7.9 Hz, H-5′), 6.8 (1H, *d*, *J* = 7.9 Hz, H-5″), 6.37 (1H, *d*, *J* = 15.9 Hz, H-8′), 6.28 (1H, *d*, *J* = 15.9 Hz, H-8″), 5.45 (1H, *ddd*, *J* = 7.5, 7.5, 3.2 Hz, H-5), 5.41 (1H, *m*, H-3), 3.99 (1H, *dd*, *J* = 7.5, 3.2 Hz, H-4), 2.36 (1H, *m*, H2-a), 2.25 (2H, *m*, H-6), 2.18 (1H, *m*, H2-b) and ^13^C-NMR (500 MHz, MeOD): 176.0 (COOH), 167.5 (C-9′), 166.9 (C-9″), 148.2 (C-4′), 148.1 (C-4″), 145.9 (C-7″), 145.6 (C-7′), 145.4 (C-3′), 145.4 (C-3″), 126.5 (C-1′), 126.4 (C-1″), 121.7 (C-6″), 121.6 (C-6′), 115.1 (C-5′), 115.1 (C-5″), 114.1 (C-8′), 113.8 (C-2′), 113.8 (C-2″), 113.7 (C-8″), 73.3 (C-1), 71.1 (C-5), 70.7 (C-3), 69.2 (C-4), 36.2 (C-6), 34.6 (C-2)

*3,4,5-Tricaffeoylquinic acid*: White powder (5 mg); ESI-MS *m/z* 677.42 [M–H]^−^ (C_34_H_30_O_15_ requires 678); ^1^H-NMR (500 MHz, MeOD): 𝛿 7.64 (1H, *d*, *J* = 15.9 Hz, H-7′), 7.55 (1H, *d*, *J* = 15.9 Hz, H-7″), 7.51 (1H, *d*, *J* = 15.9 Hz, H-7*), 7.10 (1H, *d*, *J* = 2.1 Hz, H-2′), 7.02 (1H, *d*, *J* = 2.1 Hz, H-2″), 6.99 (1H, *d*, *J* = 2.1 Hz, H-2*), 6.98 (1H, *dd*, *J* = 7.9, 2.1 Hz, H-6′), 6.92 (1H, *dd*, *J* = 7.9, 2.1 Hz, H-6″), 6.83 (1H, *dd*, *J* = 7.9, 2.1 Hz, H-6*), 6.8 (1H, *d*, *J* = 7.9 Hz, H-5′), 6.75 (1H, *d*, *J* = 7.9 Hz, H-5″), 6.70 (1H, *d*, *J* = 7.9 Hz, H-5*), 6.41 (1H, *d*, *J* = 15.9 Hz, H-8′), 6.23 (1H, *d*, *J* = 15.9 Hz, H-8″), 6.19 (1H, *d*, *J* = 15.9 Hz, H-8*), 5.77 (1H, *m*, H-5), 5.7 (1H, *m*, H-3), 5.31 (1H, *m*, H-4), 2.49 (1H, *m*, H2-a), 2.31 (2H, *m*, H-6), 2.16 (1H, *m*, H2-b) and ^13^C-NMR (500 MHz, MeOD): 174.1 (COOH), 166.9 (C-9″), 166.8 (C-9′), 167.4 (C-9*), 148.3 (C-4′), 148.25 (C-4″), 148.2 (C-4*), 146.4 (C-7*), 146. 2 (C-7″), 146.2 (C-7′), 145.4 (C-3′), 145.3 (C-3″), 145.3 (C-3*), 126.4 (C-1′), 126.2 (C-1″), 126.1 (C-1*), 121.6 (C-6″), 121.6 (C-6′), 121.6 (C-6*), 115.2 (C-5″), 115.2 (C-5*), 115.1 (C-5′), 113.8 (C-8′), 113.8 (C-2′), 113.8 (C-2″), 113.8 (C-2*), 113.5 (C-8″), 113.3 (C-8*), 72.6 (C-4), 72.5 (C-1), 69.5 (C-3), 68 (C-5), 38.7 (C-6), 35.6 (C-2)

### 4.8. Extraction for Bioassays and Quantification of CQAs

The crude hydromethanolic extracts were obtained after a methanol-water-acetic acid (75:25:2, v:v:v) maceration (200 mL) of chicory hairy roots elicitated or not, repeated three times at ambient temperature. After evaporation of solvents, the percentage of yield obtained on the basis of the dry weight (%) of crude extract was 81.6% (4.08 g of crude extract) for HR1 (not elicitated hairy roots) and 80.2% (4.02 g of crude extract) for HR2 (elicitated hairy roots). HR2 have been obtained by 9 days elicitation with 0.45 mM MeJA.

The crude extracts (3 g) were further fractionated by a liquid-liquid separation using ethyl acetate (EtOAc) and water (100 mL), three times successively. The aqueous sub-extracts were lyophilized while the ethyl acetate sub-extracts were evaporated to dryness. The yield percentage obtained on a dry weight basis (%) of each sub-extract was 69.8% (2.094 g of sub-extract) for aqueous HR1, 67.4% (2.022 g of sub-extract) for aqueous HR2; 14% (420 mg of sub-extract) for ethyl acetate HR1, and 18.1% (543 mg of sub-extract) for ethyl acetate HR2.

CQAs quantification was carried out using Prominence HPLC system (Shimadzu) consisting of a quaternary pump (LC-20AD) and a UV-visible diode-array detector (DAD; SPD-20A). Analyses were performed on a 100 × 4.6 mm Kinetex 2.6-μm PFP 100 Å column (Phenomenex, Torrance, CA, USA; http://www.phenomenex.com/). The chromatographic separation was performed using water (solvent A) and acetonitrile (solvent B), both acidified with 0.1% ortho-phosphoric acid. The solvents were delivered at a flow rate of 1.1 mL min^−1^ and the oven temperature was set at 45 °C. The following proportions of solvent B were: 5–20% (0–6 min), 20% (6–7 min), 20–25% (7–8 min), 25–5% (8–9 min), 5% (9–16 min: re-equilibration). Five μL of the extracts (12.5 mg/mL) were injected on the column. The identification of CQA and di-CQA is based on their retention time values, online UV spectra and by co-chromatography with standard samples for CQA and di-CQA. CQA and di-CQA were quantified using five-point calibration curves based on peak areas measured at 325 nm (concentration range 0.02^−1^ mg ml^−1^). Tri-CQA was quantified from the calibration curve of di-CQA, due to a lake of quantity of purified tri-CQA.

### 4.9. Antibacterial Screening of Extracts, Sub-Extracts and Pure Compounds Using Agar Dilution Method

Screening of crude hydromethanolic extracts of chicory hairy roots elicitated or not, ethyl acetate and aqueous sub-extracts, chlorogenic acid (CQA) and 3,5-dicaffeoylquinic acid (di-CQA) for antibacterial activity was carried out using clinical bacterial isolates from human samples collected in Lille (France) and from a collection of reference strains. These tests were performed using an agar dilution method on Petri dishes. The extract, sub-extracts or pure compounds dissolved in MeOH or MeOH/water were incorporated into Mueller-Hinton Agar (MHA) (Oxoid Agar, Thermofisher Scientific, Waltham, Mass., USA) at final concentrations ranging from 0.078 to 1.25 mg.mL^−1^. The final proportion of methanol in the medium did not exceed 5%, a concentration that did not affect the bacterial growth [60]. A multi-headed inoculator allowed spotting bacterial strains at 10^5^ CFU.mL^−1^ in cysteinated Ringer solution (Merck^®^, Darmstadt, Germany). The minimal inhibitory concentrations (MICs) were determined visually after 24 h of incubation at 37 °C (absence of colony on the agar surface) The same protocol was used to determine the MICs of some antibiotics (gentamicin, vancomycin and amoxicillin) and some antifungal agents (amphotericin B, fluconazole and sertaconazole). The final antimicrobial concentrations tested ranged from 0.03 to 64 µg·mL^−1^.

### 4.10. Pyocyanin and Pyoverdine Quantification Assays

To evaluate pyocyanin and pyoverdine production, *Pseudomonas aeruginosa* H103 cells untreated (DMSO 1%, *v/v*) or treated with di-CQA at concentrations ranging from 2 to 256 µg·mL^−1^ were grown in Luria-Bertani (LB) broth in a 96-well microtiter plate at 37 °C for 24 h with shaking (180 rpm). After incubation for 24 h, cell growth was determined by measuring the OD at 580 nm. A pyocyanin quantification assay was carried out as described previously [53]. Briefly, one volume of chloroform was used to extract free-cell supernatants samples. Then, ½ volume of 0.5 M HCl was added to the chloroform layer (blue layer). The absorbance of the HCl layer (red-pink layer) was recorded at 520 nm using the Spark 20M multimode microplate reader controlled by SparkControl^TM^ software Version 2.1 (Tecan Group Ltd., Männedorf, Switzerland) and the data were normalized for bacterial cell density (OD at 580 nm). To determine the production of pyoverdine, the cell-free supernatant was collected by centrifugation at 13,000 rpm for 10 min at room temperature. Then, the UV-visible spectra of pyoverdine diluted 1/10 in a 50 mM pyridine/acetic acid buffer at pH 5.0 was recorded at a wavelength ranging from 350 to 600 nm and the specific band values with a maximum absorption were normalized with cell density at OD580 nm.

### 4.11. Quantitative Biofilm Assay

To assess the propensity of *P. aeruginosa* H103 strain to form biofilms in the presence of di-CQA at concentrations ranging from 2 and 256 µg·mL^−1^, we performed crystal-violet-adhesion assays as described by O’Toole [61]. Briefly, overnight cultures were inoculated into a fresh LB broth and grown at 37 °C for 24 h in a 96-well microtiter plate under static conditions. Cell growth was determined at 580 nm. The biofilm was measured by discarding the medium, rinsing the wells with water and staining any bound cells with 0.1% crystal violet. The dye was dissolved in 30% (*w/v*) acetic acid and optical density was determined at 595 nm using the Spark 20 M multimode microplate reader controlled by SparkControl^TM^ software Version 2.1 (Tecan Group Ltd., Männedorf, Switzerland). In each experiment, the background staining was adjusted by subtracting the crystal violet bound to uninoculated controls.

### 4.12. Statistical Analyzes

To assess the significance of the differences between two groups, unpaired two-tailed t-tests were performed to calculate the *p* values using Prism GraphPad online tool (GraphPad, San Diego, CA, USA). All experiments were conducted independently with at least three replicates. The results were displayed as the mean ± standard error of the mean.

### 4.13. Antioxidant Activity Evaluation

The antioxidant activity based on the evaluation of the free radical scavenging activity of extracts of *C. intybus* L. is estimated by the DPPH method based on the assay by Brand-Williams et al. [62] with some modifications. Briefly, 1.95 mL of a freshly prepared ethanol solution of DPPH (100 μM) was mixed with 50 μL of extract or chlorogenic acid and di-CQA at various concentrations (10-350 μg.mL^−1^). The reaction solution was shaken and incubated at room temperature for 30 min in the dark and the absorbance was measured at 517 nm. In each experiment, ethanol was used as a blank. The antioxidant caffeic acid was used as a positive control. The DPPH activity of HR extracts was expressed as IC_50_ corresponding to the concentration of extract (μg.mL^−1^) required to scavenge 50% of DPPH radicals. The estimation of the IC_50_ values was done by a linear regression analysis. The EC_50_ value is IC_50_/μg DPPH and antiradical power is 1/EC_50_.

## 5. Conclusions

The present study demonstrates that HRC of *Cichorium intybus* L. (chicory) is a good alternative for the production of caffeoylquinic acid derivatives including CQA, tri-CQA and especially di-CQA. Because *C. intybus* L. is a valuable source of CQAs, this study established chicory hairy root lines under MeJA elicitation, in which the biomass and rates of CQAs are higher than *in planta* and in other HRCs explored in other species or in cultivated chicory. Concentrations of di-CQA up to 600 mg/L of HRC can be obtained in flasks, allowing chicory HRCs to be considered in bioreactors to enable large-scale production of CQA metabolites. In addition, this is the first time that tri-CQA has been identified in chicory hairy root lines under these conditions. Since, CQAs have been known for their antioxidant and antimicrobial activities [63,64], our study confirms that di-CQA displays a strong antioxidant effect and a broad-spectrum antibacterial effect, especially against strains of *Staphylococcus* and *P. aeruginosa*. In addition, the metabolite di-CQA enhanced the production of pyoverdine in *P. aeruginosa*, suggesting a link between di-CQA and iron sequestration.

## Figures and Tables

**Figure 1 antibiotics-09-00659-f001:**
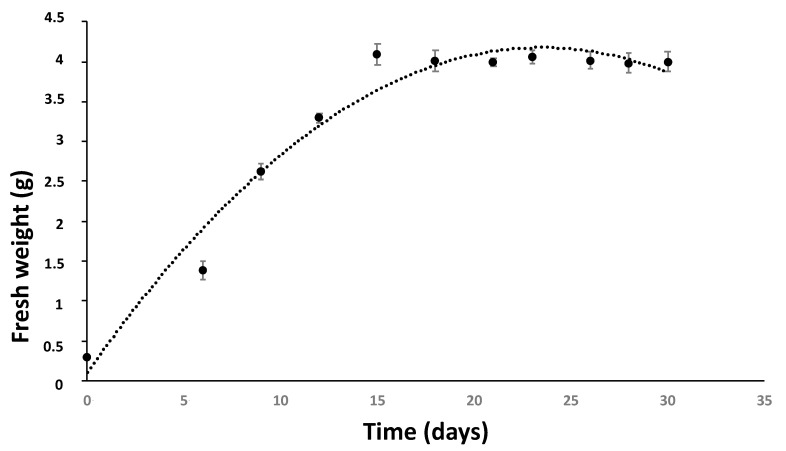
Growth kinetics of the hairy root line ‘Orchies 2659-31’ in 20 mL of liquid medium. 300 mg of roots from a 15 day-old hairy root culture were used as an inoculum and biomass production was recorded at different time points.

**Figure 2 antibiotics-09-00659-f002:**
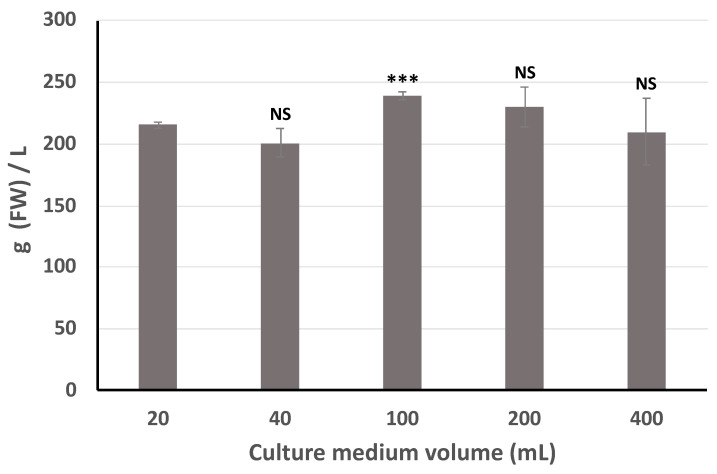
Scaling-up of chicory hairy root culture in Erlenmeyer flasks; Fresh weight (FW) was estimated on day14. Statistics were achieved by a two-tailed *t* test using Prism GraphPad online tool (https://www.graphpad.com/quickcalcs/ttest1/). The mean with SD were calculated and plotted. *** *p* = 0.0001 to 0.001; NS (Not Significant), *p* ≥ 0.05. All the conditions were compared to the 20 mL condition.

**Figure 3 antibiotics-09-00659-f003:**
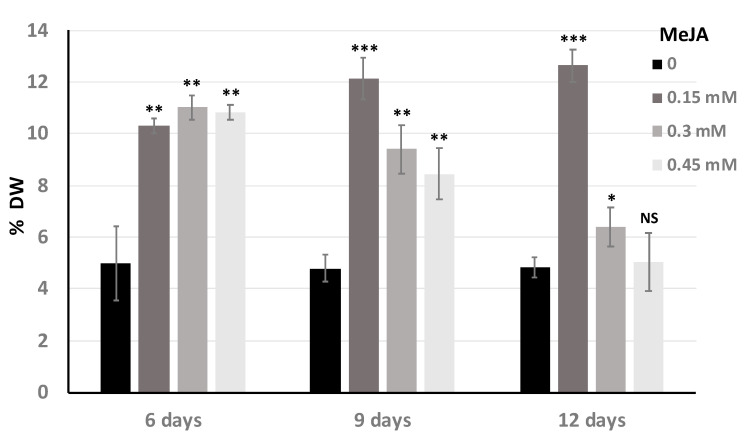
Effect of different concentrations of MeJA on the production of 3,5-dicaffeoylquinic acid (di-CQA) in the chicory hairy root line ‘Orchies 2659-31’. 0; 0.15; 0.3; 0.45 mM of MeJA indicates MeJA concentration in the culture medium. 6; 9; 12 days represents the duration of elicitation. DW: Dry weight. Statistics were achieved by a two-tailed *t* test using Prism GraphPad online tool (https://www.graphpad.com/quickcalcs/ttest1/). The mean with SD were calculated and plotted. *** *p* = 0.0001 to 0.001; ** *p* = 0.001 to 0.01; * *p* = 0.01 to 0.05; NS (Not Significant), *p* ≥ 0.05. All the conditions were compared to the control condition 0 mM MeJA.

**Figure 4 antibiotics-09-00659-f004:**
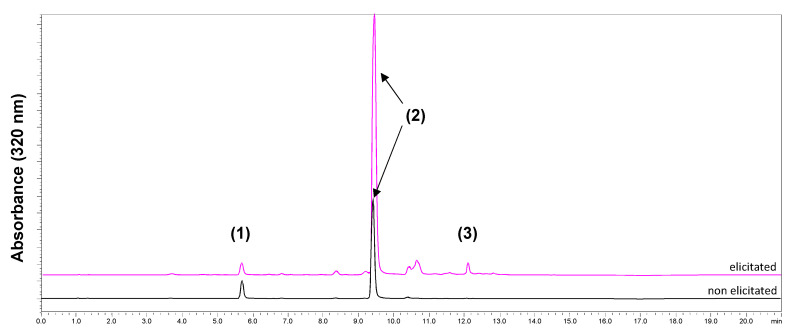
Stacked HPLC chromatograms of methanolic extracts obtained from hairy root elicitated (pink line) and not elicitated (dark line). (**1**) caffeoylquinic acid (5-CQA); (**2**) 3,5-dicaffeoyquinic acid (di-CQA); (**3**) 3,4,5-tricaffeoylquinic acid (tri-CQA).

**Figure 5 antibiotics-09-00659-f005:**
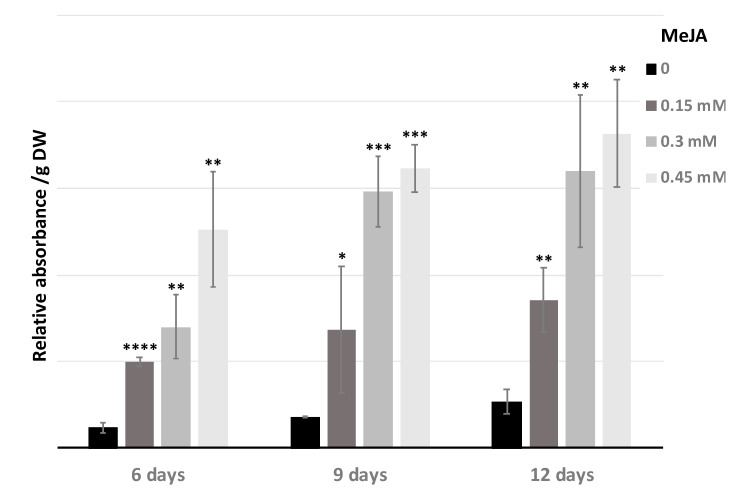
Effect of different concentrations of MeJA on the production of 3,4,5-tricaffeoylquinic acid (tri-CQA) in chicory hairy roots line. 0; 0.15; 0.3; 0.45 mM of MeJA indicates MeJA concentration of the culture medium. 6; 9; 12 days represents the duration of elicitation. DW: Dry weight. Statistics were achieved by a two-tailed *t* test using Prism GraphPad online tool (https://www.graphpad.com/quickcalcs/ttest1/). The mean with SD were calculated and plotted. **** *p* < 0.0001; *** *p* = 0.0001 to 0.001; ** *p* = 0.001 to 0.01; * *p* = 0.01 to 0.05; NS (not significant), *p* ≥ 0.05. All the conditions were compared to control condition 0 mM MeJA.

**Figure 6 antibiotics-09-00659-f006:**
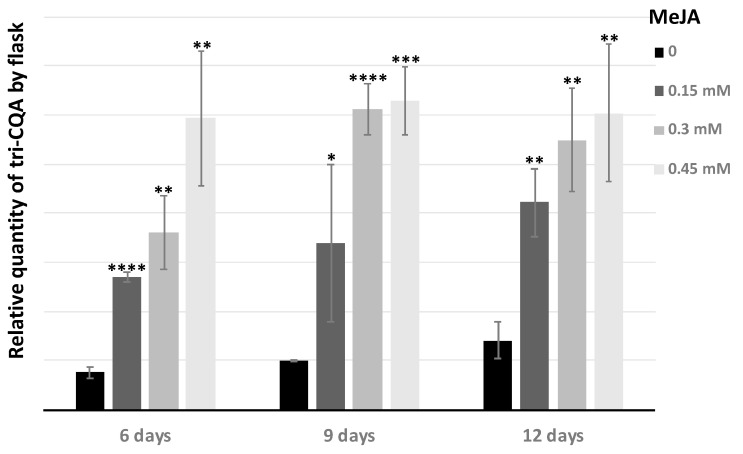
MeJA inhibitory effect on the production of 3,4,5-tricaffeoylquinic acid (tri-CQA) per flask. 0; 0.15; 0.3; 0.45 mM of MeJA indicates MeJA concentration of the culture medium. 6; 9; 12 days represents the duration of elicitation. Statistics were achieved by a two-tailed *t* test using Prism GraphPad online tool (https://www.graphpad.com/quickcalcs/ttest1/). The mean with SD were calculated and plotted. **** *p* < 0.0001; *** *p* = 0.0001 to 0.001; ** *p* = 0.001 to 0.01; * *p* = 0.01 to 0.05; NS (Not Significant), *p* ≥ 0.05. All the conditions were compared to the control condition 0 mM MeJA.

**Figure 7 antibiotics-09-00659-f007:**
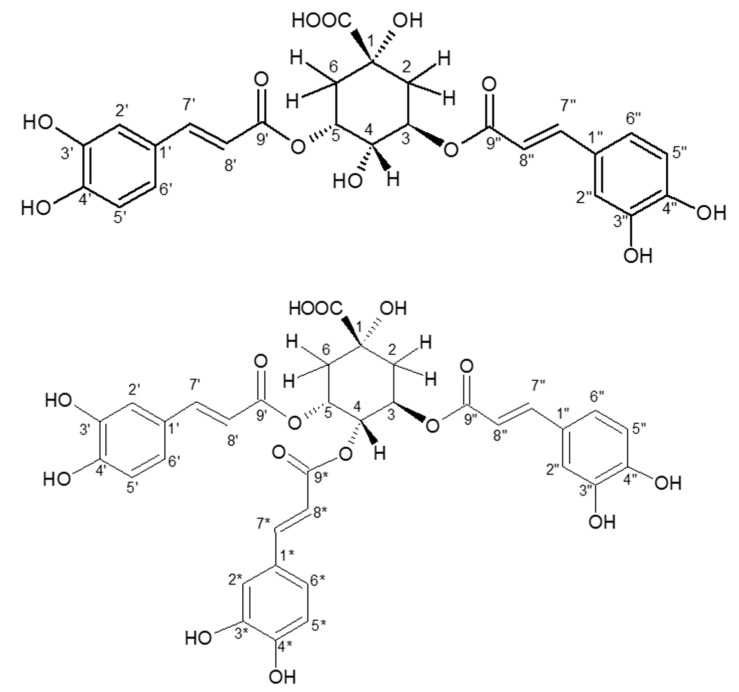
Chemical structures of 3,5-dicaffeoylquinic acid (di-CQA) and 3,4,5-tricaffeoylquinic acid (tri-CQA).

**Figure 8 antibiotics-09-00659-f008:**
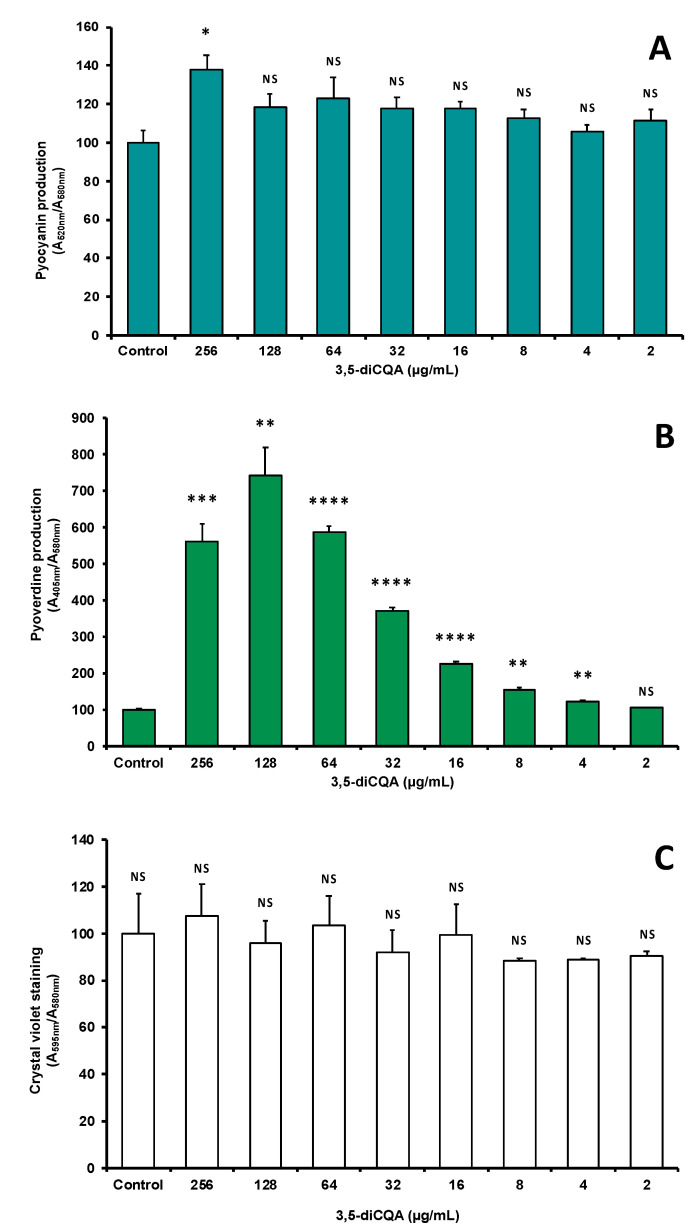
Effect of various concentrations of 3,5-dicaffeoylquinic acid (di-CQA). (**A**) On pyocyanin production by *P. aeruginosa*. (**B**) On pyoverdine production by *P. aeruginosa*. (**C**) On biofilm formation by *P. aeruginosa*. Statistics were achieved by a two-tailed *t* test using Prism GraphPad online tool (https://www.graphpad.com/quickcalcs/ttest1/). The mean with SEM were calculated and plotted. **** *p* < 0.0001; *** *p* = 0.0001 to 0.001; ** *p* = 0.001 to 0.01; * *p* = 0.01 to 0.05; NS (Not Significant), *p* ≥ 0.05.

**Table 1 antibiotics-09-00659-t001:** Quantification of CQAs (mg.g^−1^ of dry extracts) in crude extracts and sub-extracts of elicitated and non elicitated hairy root cultures.

Hairy Root Crude Extracts and Sub-Extracts		Quantification of CQAs (mg.g^−1^)		
		CQA	di-CQA	tri-CQA
**HR1**	Crude methanolic	10.01	45.01	0.3
	H_2_O	8.70	3.43	0
	EtOAc	21.80	452.15	1.58
**HR2**	Crude methanolic	10.05	69.70	1.45
	H_2_O	10.65	6.91	0
	EtOAc	18.07	508.5	8.68

HR1, non-elicitated culture; HR2, MeJA-elicitated culture; H_2_O, aqueous sub-extract; EtOAc, ethyl acetate sub-extract.

**Table 2 antibiotics-09-00659-t002:** MICs (mg.mL^−1^) of ethyl acetate sub-extracts of chicory hairy roots (EtOAc HR1, ethyl acetate sub-extract of not elicitated hairy roots; EtOAc HR2, ethyl acetate sub-extract of elicitated hairy roots) and two pure compounds (CQA, 3-caffeoylquinic acid; di-CQA; 3,5-dicaffeoylquinic acid) against some human pathogenic bacteria and fungi. MICs (in µg·mL^−1^) of some antibiotics (GEN, gentamicin; VAN, vancomycin; AMX, amoxicillin) and antifungal agents (AMB, amphotericin B; FLC, fluconazole; SER, sertaconazole) against the same pathogens panel.

Bacterial and Fungal Pathogen Strains	MIC (mg mL^−1^)				MIC (µg mL^−1^)		
	EtOAc HR1	EtOAc HR2	CQA	di-CQA	GEN	VAN	AMX
Gram positive							
*Corynebacterium striatum* T40A3	0.313	0.313	0.156	0.156	0.03	1	0.25
*Enterococcus faecalis* C159-6	NA	1.25	0.625	0.625	4	0.5	64
*Enterococcus sp.* 8153	NA	NA	NA	NA	2	4	2
*Staphylococcus aureus* 8146	0.625	0.625	0.625	0.313	0.5	2	4
*Staphylococcus aureus* 8241	0.625	0.313	0.313	0.313	0.5	2	16
*Staphylococcus aureus* ATCC 6538	0.625	0.625	0.313	0.313	0.25	2	0.125
*Staphylococcus aureus* T28-1	0.625	0.625	0.313	0.156	0.5	2	16
*Staphylococcus aureus* T17-4	0.625	0.313	0.313	0.313	0.5	2	1
*Staphylococcus warneri* T12A12	0.313	0.313	0.156	0.156	0.06	2	1
*Staphylococcus warneri* T26A1	0.313	0.313	0.156	0.156	0.06	2	0.25
*Staphylococcus epidermidis* T46A1	0.313	0.156	0.156	0.156	0.06	2	0.5
*Staphylococcus epidermidis* T19A1	0.313	0.156	0.156	0.156	32	2	8
*Staphylococcus epidermidis* T21A5	0.156	0.156	0.156	0.156	0.06	2	16
*Staphylococcus pettenkoferi* T47A6	0.156	0.156	0.156	0.156	0.06	2	0.25
*Streptococcus agalactiae* T53C9	0.625	0.313	0.313	0.156	1	0.5	0.03
*Streptococcus pyogenes* 16138	0.313	0.156	0.313	0.313	0.125	0.25	0.03
Gram negative							
*Citrobacter freundii* 11041	NA	NA	NA	1.25	0.25	>64	2
*Enterobacter aerogenes* 9004	NA	NA	NA	NA	0.5	>64	>64
*Escherichia coli* T20A1	NA	NA	NA	NA	0.25	>64	>64
*Escherichia coli* 8138	NA	NA	NA	NA	0.5	>64	>64
*Escherichia coli* 8157	NA	NA	NA	NA	0.5	>64	>64
*Escherichia coli* ATCC 25922	NA	NA	NA	NA	0.5	>64	16
*Klebsiella pneumoniae* 11016	NA	NA	NA	NA	0.25	>64	>64
*Klebsiella pneumoniae* 10270	NA	NA	NA	NA	8	>64	>64
*Proteus mirabilis* 11060	1.25	0.625	0.625	0.625	0.5	>64	2
*Proteus mirabilis* T28-3	0.625	0.625	1.25	0.313	0.5	>64	1
*Pseudomonas aeruginosa* 8131	0.313	0.625	0.625	0.625	1	>64	>64
*Pseudomonas aeruginosa* ATCC 27583	0.156	0.313	0.313	0.156	2	>64	>64
*Pseudomonas aeruginosa* 8129	0.313	0.313	0.313	0.313	0.03	>64	>64
*Salmonella sp.* 11033	NA	NA	NA	NA	0.25	>64	2
Fungi					**AMB**	**FLC**	**SER**
*Candida albicans* 10286	NA	0.625	NA	1.25	4	32	>64
*Candida albicans* ATCC 10231	0.156	0.156	0.078	0.078	0.5	8	64

NA, non-active (MIC value ≥ 1.25 mg.mL^−1^).

**Table 3 antibiotics-09-00659-t003:** DPPH radical scavenging activities of the different *Cichorium intybus* L. (chicory) hairy root preparations and caffeoylquinic acid derivative compounds.

Samples	IC_50_ (µM)	50 (µM/µmol DPPH)	Antiradical Power (nM/µmol DPPH)	IC_50_ (µg/mL)	EC_50_ (µg/mL/µg DPPH)	Antiradical Power (µg/mL/µg DPPH)
crude methanolic extract HR1	-	-	-	89.96	1.17	0.86
ethyl acetate sub-extract HR1	-	-	-	24.06	0.31	3.20
aqueous sub-extract HR1	-	-	-	345.32	4.49	0.22
crude methanolic extract HR2	-	-	-	52.76	0.69	1.46
ethyl acetate sub-extract HR2	-	-	-	18.40	0.24	4.18
aqueous sub-extract HR2	-	-	-	217.90	2.83	0.35
3-caffeoylquinic acid (CQA)	37.70	193.32	5.17	13.35	0.17	5.76
3,5-dicaffeoylquinic acid (di-CQA)	16.03	82.20	12.17	8.28	0.11	9.29

HR1, non-elicitated culture; HR2, MeJA-elicitated culture. IC, inhibition concentration; EC, effective concentration.

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
