# Peer review of "MeJA Elicitation of Chicory Hairy Roots Promotes Efficient Increase of 3,5-diCQA Accumulation, a Potent Antioxidant and Antibacterial Molecule"

_antibiotics, 2020, doi:10.3390/antibiotics9100659_

Round 1
Reviewer 1 Report
This is an interesting paper, in which four separate part could be seen: (i) biotechnological procedurÄ™ for the production of caffeoylquinic acids; (ii) quite standard studies on antibacterial activity of various extracts of chicory hairy roots growth medium; (iii) influence of 3,5-dicaffeoylquinic acid on biofilm formation by Pseudomonas aeruginosa; studies of antioxidant activity of the isolated acids, which are rather obvious. The most important are biotechnological studies, especially finding that the use of plant hormone increases tha productivity of 3,5-dicaffeoylquinic acid and provides three-substituted one. Studies on biofilm formation of P. aeruginosa are interesting in that 3,5-dicaffeoylquinic acid selectively blocks production of siderofore – pyoverdine. However, without influence on biofilm formation. I consider this part as some preliminary studies on the possible mechanism of antibacterial action of 3,5-dicaffeoylquinic acid. In my opinion the lower productivity of pyoverdine do not result from coompetition for Fe(III) ions because siderophores are far better complexones, but Authors mention this in a soft, safe way as „that this compound would interact wit Fe3+” .
The weak part of the manuscript is its title, which concentrates on the third part of the studies. In my opinio nit should be changed in such a way that it will point out biotechnological aspect of the production of caffeoylquinic acids.
Reviewer 2 Report
The manuscript for the production of CQA derivatives is interesting. The research results presented therein are of great practical importance.
However, I have a few comments.
In my opinion, chromatographs (Figures 7 and 9) should be included in the Supplementary Information. In the manuscript, the mere description of the results is sufficient.
Point 2.5
The fragment of the text contained in lines 237-242 should be corrected as it contradicts the data presented in Figure 9.
Point 4.6
Line 472 - What were the proportions of ethyl acetals and water.
Minor remarks
Figure 6 - it should be days, not jours
In Materials and Methods, Latin names should be italicized, for example Cichorium intybus or Rhizobium rhizogenes.
Reviewer 3 Report
The Manuscript is well organized and the results are clearly reported. The main critical spect is the lack of a quantitative evaluation of the main metabolites (CQAs) in the samples used for themicrobiological test. I suggest to improve this part adding new informations on this aspect.
More detailed comments are in the attacched file

Round 2
Reviewer 3 Report
I confirm the work is well organized and clearly discussed, so I think it is suitable for publication. Only one aspect remains to be changed: the way in which quantitative data are expressed in Table 1.
The last column of the table reads "Quantification of CQAs (% MS)". What does% MS mean? The authors use this acronym to define Murashige Skoog medium (lines 99 and 413).
At lines 567-568 (section 5 4.8), the authors correctly reported: "CQA and di-CQA were quantified using five-point calibration curves based on peak areas measured at 325 nm (concentration range 0.02-1 mg mL - 1) ". Because the dry weights of the different extracts have been determined and the values are reported at lines 548-557, I was waiting to see the results expressed in mg / g of dry extract or micrograms / g of dry extract.
Please clarify this point
